# Current Approaches in Surgical and Immunotherapy-Based Management of Renal Cell Carcinoma with Tumor Thrombus

**DOI:** 10.3390/biomedicines11010204

**Published:** 2023-01-13

**Authors:** Marina M. Tabbara, Javier González, Melanie Martucci, Gaetano Ciancio

**Affiliations:** 1Department of Surgery, Miami Transplant Institute, University of Miami Miller School of Medicine, Jackson Memorial Hospital, Miami, FL 33136, USA; 2Servicio de Urología, Unidad de Transplante Renal, Hospital General Universitario Gregorio Marañón, 28007 Madrid, Spain; 3Miami Transplant Institute, University of Miami Miller School of Medicine, Jackson Memorial Hospital, Miami, FL 33136, USA; 4Department of Surgery and Urology, Miami Transplant Institute, University of Miami Miller School of Medicine, Jackson Memorial Hospital, Miami, FL 33136, USA

**Keywords:** renal cell carcinoma, tumor thrombus, pathology, nephrectomy, thrombectomy, multimodal treatment, immunotherapy

## Abstract

Renal cell carcinoma (RCC) accounts for 2–3% of all malignant disease in adults, with 30% of RCC diagnosed at locally advanced or metastatic stages of disease. A form of locally advanced disease is the tumor thrombus (TT), which commonly grows from the intrarenal veins, through the main renal vein, and up the inferior vena cava (IVC), and rarely, into the right cardiac chambers. Advances in all areas of medicine have allowed increased understanding of the underlying biology of these tumors and improved preoperative staging. Although the development of several novel system agents, including several clinical trials utilizing immune checkpoint inhibitors and combination therapies, has been shown to lower perioperative morbidity and increase post-operative recurrence-free and progression-free survival, surgery remains the mainstay of therapy to achieve a cure. In this review, we provide a description of specific surgical approaches and techniques used to minimize intra- and post-operative complications during radical nephrectomy and tumor thrombectomy of RCC with TT extension of various levels. Additionally, we provide an in-depth review of the major developments in neoadjuvant and adjuvant immunotherapy-based treatment and the impact of ongoing and recently completed clinical trials on the surgical treatment of advanced RCC.

## 1. Introduction

Renal cell carcinoma (RCC) accounts for 2–3% of all malignant disease in adults, with 30% of RCC diagnosed at locally advanced or metastatic stages of disease [1,2]. Neoplastic extension of the primary tumor into vascular structures forms a tumor thrombus (TT), which grows into the main renal vein and up the inferior vena cava (IVC) in up to 10% of cases of RCC. In rare cases, the TT can grow into the right cardiac chamber in about 1% of cases [1,2,3,4,5,6,7].

Historically, chemo- or radiotherapy protocols had been deemed ineffective in attaining complete control the tumor burden of advanced RCC. Surgical monotherapy provided remission in organ-confined tumors, while variable outcomes occurred in cases where lymphadenectomy, tumor thrombectomy, and/or metastasectomy were needed alongside primary tumor excision [8,9]. Neoadjuvant therapy aims to eradicate micro-metastatic disease and reduce surgical complexity [10]. Advances in immunotherapy with interleukin-2 (IL-2) and interferon-alpha (IFN-α) have been shown to improve the overall survival in patients with late-stage disease [8,9,11,12]. Vascular endothelial growth factor (VEGF) pathway inhibitors and mammalian target of rapamycin-inhibitors have provided significant clinical benefits for patients with advanced stage RCC [13]. More recently, immune checkpoint inhibitors (ICI) and combination therapies have revolutionized the management of advanced RCC, showing additional progression-free and survival benefits [14,15].

Although the development of several novel agents for the treatment of metastatic RCC has been shown to lower perioperative morbidity and increase post-operative recurrence-free and progression-free survival, surgery remains the mainstay of therapy to achieve a cure. However, complications can often be devastating, making the procedure a challenge for both the surgeon and patient. The use of liver transplant-based approaches during open radical nephrectomy and tumor thrombectomy has permitted the safe and effective surgical management of RCC with TT without necessitating the use of cardiopulmonary bypass (CPB) [16]. Minimally invasive surgery utilizing laparoscopic and robotic techniques for the management of radical nephrectomy and IVC tumor thrombectomy has been increasingly reported in the literature [17,18,19,20,21,22]; however, its role in the setting of RCC with TT is of debate and requires further investigation [23].

The aim of this review is to provide a detailed description of specific surgical approaches and techniques used to minimize intra- and post-operative complications during radical nephrectomy and tumor thrombectomy of RCC with TT extension. We describe the necessary preoperative workup and the use of liver transplant-based surgical techniques that can be successfully applied for resection of high-level tumor thrombi. Additionally, we discuss in detail the major developments in neoadjuvant and adjuvant immunotherapy-based treatment of advanced RCC, including a review of ongoing and recently completed clinical trials.

## 2. Surgical Approaches

### 2.1. Preoperative Planning and Imaging

Surgical treatment of patients with RCC with venous TT requires meticulous preoperative planning. Advances in imaging has facilitated precise delineation of primary tumors, the detection of lymphadenopathy, and intra-abdominal metastasis [5,24]. MRI can also be utilized to determine the extent of TT and the degree of IVC occlusion [4,7]. Further preoperative assessment with transesophageal (TEE) can be considered if the TT is shown to extend into the right atrium [7,25].

One of the most crucial elements in preoperative workup is to determine the level of the tumor thrombus [25]. The Neves-Zinke classification [26] is one of the most common classification systems used to define the level of the thrombus. A modified approach for classifying level III thrombi (retrohepatic) has been described by authors at the University of Miami [27]. Level III thrombi are categorized into four groups:IIIa (retrohepatic IVC below major hepatic veins)IIIb (retrohepatic IVC reaching the ostia of major hepatic veins)IIIc (retrohepatic IVC and extending above major hepatic veins, but below diaphragm)IIId (suprahepatic and supradiaphragmatic IVC, reaching intrapericardial IVC, but infra-atrial)

### 2.2. Radical Nephrectomy and Tumor Thrombectomy

#### 2.2.1. Level I TT

For patients with RCC with level I thrombi, the principles of traditional radical nephrectomy can be applied [28]. An open approach for radical nephrectomy with tumor thrombectomy has been performed using a midline, chevron, or subcostal incision, providing great exposure of bilateral renal hila [5,29]. The kidney can be mobilized medially until reaching the renal artery outside the Zuckerkandl fascia, where the artery is ligated and divided [25]. Early ligation of the renal artery allows for collapse of collateral circulation, reducing bleeding and facilitating further dissection [27].

#### 2.2.2. Level II TT

In case of level II thrombi, it is crucial to obtain adequate exposure and control of the infrahepatic and retrohepatic IVC before cavotomy and thrombectomy [25,30]. This can be achieved through mobilization of the posterior surface of the liver. Some small hepatic and lumbar veins should then be ligated and divided. Vascular clamps can then be placed on the contralateral renal vein and IVC below and above the thrombus. Then, a cavotomy and thrombectomy can be performed. Clamping below the hepatic venous confluence obviates the need for bypass due to collateral venous return via the lumbar, azygos-hemiazygos, and portal venous systems [5].

#### 2.2.3. Level III and Level IV TT

Level III thrombi necessitate characterization of the tumor level on preoperative imaging and the use of intraoperative TEE [31]. Although successful use of cardiopulmonary bypass (CPB) with circulatory arrest has been reported in the resection of high-level TT [30], there remains a risk of coagulopathy, platelet dysfunction, and central nervous system complications [25,32]. In efforts to avoid CPB, complete intraabdominal resection of high-level thrombi has been achieved with the use of hepatic mobilization maneuvers [31,32,33,34]. Ciancio et al. [16,33] have described their transplant-based approach at length. First, the liver is mobilized and rolled to the left abdomen. Then, they gain surgical control of the hepatic hilum, which permits isolation and control of the porta hepatis. This allows for the use of the Pringle maneuver if the thrombus extends above the hepatic veins [35,36]. Afterwards, a Piggyback maneuver can be performed [37]. Total circumferential dissection of the IVC is obtained via ligation and division of minor hepatic veins draining into the anterior surface of the IVC, as well as dissection of posterior surface or the IVC (Figure 1).

A clamp may be applied on IVC below the hepatic venous outflow if the TT can be milked down below the hepatic venous confluence. This avoids liver congestion [38]. To assess the level of the clamp and the potential dislodgement of the thrombus and possible pulmonary embolism, TEE monitoring should be utilized. For level IIId and for those cases where the milking maneuver is not feasible, dissection continues until the supradiaphragmatic and intrapericardial IVC is exposed. The central tendon of the diaphragm requires opening in the midline for intrapericadial IVC exposure. After complete circumferential control of the IVC is obtained, gentle traction at the cavo-atrial junction permits the relocation of the right atrium inside the abdomen [39]. Then, sequential vascular clamping should be performed in the following order: (1) IVC below the thrombus, (2) contralateral renal vein (and right adrenal vein in case of left-sided renal tumor), (3) Pringle maneuver, and (4) IVC above the TT (below the MHVs if milking maneuver was successful or supradiaphragmatic IVC if not). Afterwards, thrombectomy is performed (Figure 2).

Right atrium control may be gained exclusively through the abdomen following the principles of the transplant-based approach already mentioned. This avoids the need for CPB in most instances [39].

## 3. Systemic Therapy in Locally Advanced RCC

While radical nephrectomy with tumor thrombectomy remains the gold standard treatment for advanced RCC, the use of immunotherapy agents for RCC with high-level venous TT involvement in the neoadjuvant and adjuvant setting has been increasingly discussed in the literature [10,40,41]. Several clinical trials have been recently published with promising outcomes; however, there remains a lack of consensus in the literature regarding the effectiveness, safety, and clinical utility of immunotherapy in the treatment of advanced RCC [42,43,44,45].

### 3.1. Neoadjuvant Treatment in Locally Advanced RCC

#### 3.1.1. Rationale for Neoadjuvant Treatment Use

Neoadjuvant therapy was initially added to the basic treatment schedule in RCC with the aim to achieve a reduction in disease burden prior to surgical resection, to simplify the surgical approach in those cases considered more complex, and to select patients with an adequate response to systemic therapy who could benefit from such surgical debulking [46,47,48,49,50,51]. In the context of locally advanced disease, neoadjuvant treatment could also provide a greater likelihood of achieving a complete resection in those high-risk cases with intimate attachment or invasion of adjacent organs/large retroperitoneal vessels, which would require complex maneuvers, extensive resection, and subsequent difficult reconstructions. Finally, the concept of neoadjuvant treatment has also been extended to cases in which nephron-sparing surgery was highly recommended or mandatory [52].

To analyze the response to neoadjuvant treatment, the images taken from the staging study performed prior to the start of systemic treatment are used, and further compared with the images obtained from a new study that should be performed at the end of neoadjuvant treatment and before proceeding to surgical resection. In this way, certain measurements from both studies can be compared providing in turn an objective response rate. These measurements include the reduction in tumor size according to RECIST criteria and the decrease in tumor complexity according to the RENAL nephrometry score [53,54]. This allows subjective judgments to be made about the difficulty in adapting to the established resection plan or facilitating the performance of nephron-sparing surgery. Treatment-related toxicity and postoperative complications are usually recorded following the recommendations established in the NCI CTCAE v3.0 and Clavien-Dindo classifications, respectively [55,56].

#### 3.1.2. Obtaining Complete Resection in the Locally Advanced Disease Setting

The use of neoadjuvant treatment with the aim to obtain a complete/almost complete resection in the setting of locally advanced RCC is circumscribed to date to sunitinib, sorafenib, and axitinib. A summary of published trials in neoadjuvant therapy in locally advanced renal cell carcinoma is depicted in Table 1.

The first study to evaluate the feasibility and efficacy of neoadjuvant therapy prior to surgical treatment in the setting of locally advanced disease was conducted by Thomas et al. [52]. Their work included a total of 19 patients who had extensive local disease that was considered unresectable. Patients were administered sunitinib 50 mg/day for a total 4-week course. Initial analysis showed a partial response in 3 of the patients (16%) with a median reduction of 24%. This reduction allowed intervention in 4 patients (21%). However, the authors reported a non-negligible severe toxicity rate (grades 3–4) of 37%. The report notes an absence of unexpected surgical morbidity, but no data are provided on major complications following the surgical procedure. A second study by Hellenthal et al. [57] also tested sunitinib (in this case, at a dose of 37.5 mg/day for 3 months) in a total of 20 patients (16 of whom had locally advanced disease). Overall, 85% of patients experienced a reduction in tumor size (median 11%). After neoadjuvant treatment, all patients underwent laparoscopic partial or radical nephrectomy. The most important toxicities attributed to the use of sunitinib were gastrointestinal (65%) and hematologic (55%). Finally, the update of the phase-II trial with sunitinib 50 mg/day in two 6-week cycles conducted by Rini et al. [58], included data on a total of 28 patients with unresectable primary tumors. The authors noted that after neoadjuvant treatment, the disease volume was reduced by 22%, and that in 13 patients (45%), nephrectomy could be finally attempted with success.

Cowey et al. [59] conducted a phase-II trial including a total of 30 patients (17 of whom harboring locally advanced disease). All of them received neoadjuvant treatment with sorafenib (400 mg twice daily) for approximately 1 month (median 33 days). They experienced a reduction in disease volume of 9.6%, and disease stabilization according to RECIST criteria was obtained in 86.6% during treatment administration. All patients subsequently underwent nephrectomy without experiencing surgical complications attributable to the use of sorafenib. On the other hand, Hatiboglu et al. [60] reported their experience in a total of 12 patients in a double-blind, placebo-controlled trial designed to test the downsizing effect of sorafenib prior to surgical treatment in patients with locally advanced RCC. Patients were randomized 3:1 to sorafenib (400 mg/2 times daily for 4 weeks) or placebo. The group treated with sorafenib presented a significant reduction (*p* < 0.05) in tumor size in 29% of the cases, which allowed the intervention to be performed in a total of 9/12 patients (4/12 partial and 5/12 radical nephrectomy, respectively).

The experience with axitinib was reported by Karam et al. [61] in a total of 24 patients with biopsy-proven conventional T3a RCC. Axitinib neoadjuvant treatment was prolonged for 12 weeks. The primary objective of this study was to demonstrate a reduction in disease volume according to RECIST criteria. Overall, they observed a reduction in tumor size of 28.3%, achieving partial responses in 11 patients (46%) according to the established standard. However, the toxicity rate in this series was considerable, showing 52% grade 2 toxicity and 8% grade 3 toxicity. Reported toxicity included hypertension, fatigue, mucositis, hypothyroidism, and hand-foot syndrome. Likewise, the authors reported a rate of major complications (Clavien 3–5) of 12.5%, which corresponded mainly to chylous ascites (2 cases) and hemorrhage that required relaparotomy (1 case).

#### 3.1.3. Facilitating Conservative Surgical Treatment

An additional theoretical advantage of neoadjuvant treatment is facilitating nephron-sparing surgery. The first study in this regard was carried out by Silberstein et al. [53] in 2010 with sunitinib (50 mg/day for 12 weeks). The pilot study included a total of 14 tumors with an imperative indication for conservative surgery. The observed reduction in tumor size was 21.1% (7.1–5.6 cm). Although in only four of the cases (28.6%) a partial response according to the RECIST criteria was achieved, conservative surgery could be carried in all the sample, without positive margins or the need for permanent dialysis after the intervention. However, in three of the cases (21.4%) a urine leak was observed that required conservative management for its resolution.

Shortly after, Lane et al. [62] retrospectively collected the multi-institutional experience regarding the effects of sunitinib in tumor downsizing to allow the performance of a partial nephrectomy in a series that included *n* = 78 cT1-cT3 RCCs that presented a median size of 7.2 cm. Patients were randomized to 37.5 mg or 50 mg sunitinib daily for 6 weeks. The authors observed an objective reduction of 32%, with 15 partial responses and improvement in the RENAL score in up to 59% of the cases. Grade 3 toxicity reported was 14%. Negative predictors of tumor size reduction were the presence of visible lymph nodes on imaging tests, the presence of a histological variant other than the conventional one, and the presence of high grade on pathological analysis. Surgery could be performed in 68 cases (87%), in 49 of which a partial nephrectomy was achieved (63%). The authors reported a postoperative complication rate of 7% that included urine leakage, arteriovenous fistula, incisional hernia, and the need for permanent postoperative dialysis among other causes. Of note, the authors conclude that neoadjuvant treatment with sunitinib leads to objective debulking in most cases, which would eventually benefit from conservative surgical treatment with acceptable morbidity.

McDonald et al. [54] retrospectively reviewed the experience with sunitinib (50 mg/day for 12 weeks) regarding the functional results observed in patients with complex renal masses and indication for imperative partial nephrectomy (previously considered unfeasible). The study included 47 patients (mean tumor size 7.2 cm and mean RENAL score 11) in the neoadjuvant treatment group, and 78 patients (mean tumor size 6 cm, and mean RENAL score 10). The treatment group received 50 mg sunitinib/day for 12 weeks. The reduction in size and RENAL score observed in the group on sunitinib reached 5.8 cm (*p* = 0.012) and 9 (*p* = 0.001), respectively. High-grade toxicity (3–4) was recorded in 29.8% of the patients in the treatment group, although no significant differences were observed in the rate of major surgical complications observed after the intervention between both groups. The mean glomerular filtration rate (GFR) observed after the intervention did not show any significant difference between both groups (*p* = 0.53). Thus, the authors conclude that neoadjuvant sunitinib at a dose of 50 mg/day for 12 weeks can facilitate the surgical treatment of a theoretical patient harboring a complex renal mass in whom indication for partial nephrectomy is mandatory, without compromising the renal function after the intervention.

In addition to sunitinib, axitinib and pazopanib have also been tested in this context. Karam et al. [61] examined the response to axitinib in a phase-II trial that included patients with T3a RCC. Among all the 24 patients included, only 5 (22%) received a partial nephrectomy. No procedure-related postoperative complications were observed in any of the patients who underwent partial surgery. The AXIPAN study (phase-II trial) [63] was designed to test the response regarding tumor size to neoadjuvant axitinib on T2 RCC patients that would subsequently facilitate the performance of a partial nephrectomy. Varying doses of axitinib 5–10 mg twice daily were administered for 2–6 months prior to surgical treatment. The primary outcome of the study was to determine the percentage of patients who would receive partial nephrectomy treatment for a tumor > 7 cm after neoadjuvant treatment with axitinib. A total of 18 patients with a mean tumor size of 7.7 cm were included. Sixteen of them (89%) experienced a reduction in tumor size (up to 6.4 cm; objective reduction of 17%; *p* = 0.001). All patients who experienced a sufficient reduction according to the established criteria underwent partial nephrectomy, and the authors considered that the success contemplated in the primary outcome had been achieved in a total of 12 patients (67%). It should be noted that 27.8% of the patients included in the study presented grade 3 toxicity with the drug and, while up to 27.8% of experienced a Clavien-Dindo grade III-V complication after the intervention. The authors concluded that the benefit obtained with axitinib in neoadjuvant therapy was modest at best and potentially required significant expertise to avoid the occurrence of relevant postoperative complications after the procedure.

Finally, Rini et al. [64] tested the possibility of using pazopanib as a first-line neoadjuvant agent aiming to facilitate conservative surgery in patients with limited renal function (mean GFR 54 mL/min/1.73 m^2^) and presence of complex renal masses (mean RENAL score 11). Of note, up to 56% of the 25 patients included had a solitary kidney. Patients received pazopanib 800 mg/day for 16 weeks, with a dose reduction to 600–400 mg in the event of intolerable toxicity during administration of the first treatment cycle. The authors observed a significant reduction (*p* < 0.001) in tumor size and volume (from 7.2 to 5.5 cm and 170 to 92 cc, respectively). However, after the intervention, 5 patients (20%) who underwent partial nephrectomy presented relevant postoperative complications (including postoperative fistula) and required permanent dialysis after the procedure.

It can be concluded that neoadjuvant treatment results in a modest, yet significant reduction in tumor size and volume that could facilitate conservative surgery in selected cases with acceptable morbidity and functional outcomes. However, the absence of comparative studies of sufficient quality currently prevents the universal adoption of this type of treatment for this purpose.

#### 3.1.4. Downstaging IVC Tumor Thrombus Level

The role of neoadjuvant treatment with targeted agents has also been investigated in downstaging of tumor thrombus within the inferior vena cava (IVC). Three different studies collect the experience in this regard. The first study was conducted by Cost et al. [65]. They included 25 patients. Almost half of these patients (48%) received sunitinib, while the remaining sample received other alternative target therapies. Overall, a decrease in tumor thrombus height was observed in 44% of the cases, while the height of the thrombus inside the IVC did not change or increased in the remaining cases. When sunitinib-treatment group was analyzed separately, 84% of those patients treated experienced no change in tumor thrombus height, while only in 12% of the cases a reduction in thrombus height was recorded. The conclusion proposed by the authors identifies a very modest clinical effect (hardly measurable) in downstaging of tumor thrombus, which is only perceived in cases treated with sunitinib and not with the other agents tested. Similarly, Bigot et al. [66] randomized 14 patients with RCC in conjunction with tumor thrombus to receive sunitinib or sorafenib. The results obtained showed no change in the height of the tumor thrombus in most of the treated patients (84%) with significant presence of grade 3 toxicity (21%). In contrast, a study carried out by Zhang et al. [67] including 18 patients that received 400 mg of sorafenib twice a day, showed a reduction in the height of the tumor thrombus in 4 of them with acceptable toxicity (>grade 3) and no evidence of healing troubles. Lastly, Field et al. [68] performed a retrospective evaluation of the multi-institutional experience on a total of 53 patients with RCC and vascular involvement. The effect of neoadjuvant treatment was compared between those who received sunitinib and those who did not receive neoadjuvant treatment. Partial response was observed in 28% of patients. A decrease in thrombus height was observed in 52% of treated patients (median decrease 1.3 cm), which did not translate into a significant benefit in overall survival.

#### 3.1.5. Ongoing and Recently Completed Trials

There are currently 13 ongoing or recently completed trials for M0 localized or locally advanced RCC, including 1 pilot study, 2 phase-I studies, and 10 phase-II studies. Only one of the studies includes overall survival as the primary outcome variable, the remaining trials contemplate disease-free survival, recurrence-free survival, or progression-free survival as main outcome variables. Like previous studies, the majority of the trials included mainly patients with clear cell RCC. Not surprisingly, most of the trials have shifted from TKIs in monotherapy (the study conducted by Karam et al. [61] with axitinib and Abdelaziz et al. [69] with cabozantinib) to immunological checkpoint inhibitors (ICIs). It seems that most groups have accepted this change as a new paradigm in the treatment of locally advanced disease. Several of these new studies are testing nivolumab in monotherapy. Notably, the PROSPER study (phase-III multicentric study) plans the enrollment of 766 patients aiming to test nivolumab in monotherapy in the neoadjuvant setting. Although the study allows the inclusion of M1 patients, there must be a period of “no evidence of disease” achieved by metastasectomy, ablative treatment, or stereotactic radiotherapy for 12 weeks prior to the start of systemic treatment. This study will end the enrollment of patients in the late 2023. Most of the studies that combine TKIs with other ICIs are still ongoing, such as Keynote-564 [50], Checkmate-914 [44], IMmotion151 [70], and include additional correlative studies to determine the true effect of microenvironment and to seek certain biomarkers that maybe of potential interest to anticipate the tumor response to treatment [71,72].

A post-hoc analysis of CheckMate 214 trial [73], nivolumab plus ipilimumab, demonstrated superior efficacy over sunitinib in patients with previously untreated International Metastatic RCC Database Consortium (IMDC) intermediate/poor-risk advanced RCC. The trial showed improved overall survival and progression-free survival benefits with nivolumab plus ipilimumab (*n* = 60) vs. sunitinib (*n* = 52) in intermediate/poor-risk patients with sarcomatoid features. A higher proportion of patients achieved an objective response (56.7% vs. 19.2%) and a higher proportion of patients achieved a complete response (18.3% vs. 0%).

### 3.2. Adjuvant Treatment in Locally Advanced Disease

Some clinical trials have been reported testing sunitinib (S-TRAC, ASSURE), sorafenib (ASSURE), and pazopanib (PROTECT) in this context [72,74,75]. Only S-TRAC showed an increase in the disease-free period, but none of the available clinical trials were able to demonstrate any beneficial effect on overall survival. The results obtained in S-TRAC allowed the approval of sunitinib by the FDA for the treatment of these patients. The European Medicines Agency (EMA) has not yet approved the use of any of these adjuvant agents in the M0 context due to a lack of balance between the risk assumed and the potential benefit. Pool analysis of data from all of these trials demonstrated no benefit in terms of disease-free or overall survival among patients given any of the above agents if surgical treatment achieved complete resection of neoplastic tissue [76]. The improvement in terms of disease-free survival is likely to be more feasible with full-dose protocols and in high-risk patients. In contrast, full-dose protocols were associated with an increase in the frequency and severity of adverse effects. Therefore, adjuvant protocols in the patient who does not have metastases should be considered experimental and should not be performed outside the scope of a clinical trial. Its use to try to reduce the height of the tumor thrombus inside the inferior vena cava should not be recommended. Rather, adjuvant treatment should be recommended in the context of a multimodal protocol in the M1 patient, in whom metastatic spread cannot be completely controlled with local treatment. The recommendations for this type of patient are the same as for any patient with metastatic disease (M+) and should be carried out according to risk stratification. The ideal time to start systemic treatment has yet to be precisely defined, since some RCCs present a relatively indolent course for a variable period. During this period, observation could be an adequate alternative, if the patient presents few symptoms and/or limited tumor burden [77,78,79].

## 4. Conclusions

Surgery remains the gold standard of treatment in RCC with vascular involvement, obtaining more than acceptable survival rates in the best candidates. Advances in all areas of medicine have allowed an increase in the understanding of the underlying biology of these tumors and a better preoperative staging. The routine use of cross-sectional imaging techniques in the diagnosis of these masses has facilitated the identification of the anatomical relationships established between the cranial end of the TT and the rest of the surrounding structures, which in turn has allowed the establishment of classifications based on the anticipation of surgical maneuvers necessary for their correct intraoperative management. In addition, the use of TEE has increased safety in the treatment of those most proximal thrombi by offering a dynamic diagnosis in real time during the performance of the maneuvers necessary to control its cranial end when it is close or inside the right atrium.

Adequate surgical planning should contemplate aspects related to the incision, the use of an adequate auto-static retractor, and the set and sequence of maneuvers necessary to establish optimal and safe vascular control for the patient. While in more distal thrombi, IVC control is relatively simple, in more distal TTs (III-IV) it is possible that techniques derived from transplantation offer a safer approach for the patient. Hepatic mobilization, Pringle maneuver, circumferential dissection of the IVC, two-step cavotomy or descent and abdominalization of the right atrium may be necessary depending on the level reached by the TT inside the IVC. In those cases with complete or near-complete obstruction of the IVC and adequate collateral venous circulation, cavectomy should be considered as a path to simplify the approach, reduce the risk of complications, and improve the rate of surgical margins at minimal cost to the patient. Although cardiopulmonary bypass continues to represent an option for large volume intracardiac thrombi, its use can be obviated by employing the aforementioned maneuvers in most cases.

Published studies with VEGFi tyrosine kinase inhibitors in the neoadyuvant setting have shown modest responses for locally advanced RCC. However, the only four prospective trials carried out in this context are of small volume (pilot scale or phase-II), use VEGFi in monotherapy, and have been applied mainly to patients with clear cell variant who presented high heterogeneity in the cohorts, therapeutic agent, primary outcome variable, and design. The outcome variable used by three of these four studies was reduction in tumor volume or surgical complexity (that allowed the performance of partial nephrectomy in patients with limited renal function or in whom the surgeon considered it complex to be performed). The duration of neoadjuvant treatment varied considerably between studies (from 4 weeks to 6 months), as did the mean waiting time between completion of neoadjuvant treatment and intervention (of 36 h to >7 days, or at the discretion of the consensus between the oncologist and the surgeon). The objective response rate observed ranged only from 22–46%. Pragmatically, it is difficult to extrapolate the results of these studies to clinical practice given the small size of the samples, the use of different agents without a control arm, and the implicit selection bias in two of the studies, in which positive discrimination was performed by including only poor candidates for partial nephrectomy.

In addition to the lack of sufficient evidence about the real benefit in terms of survival shown by these agents in these trials, the problem of medication-related side effects (effects related to postoperative recovery and increased incidence and severity of surgical complications) remains, as well as the potential deleterious effects derived from the delay in the intervention, particularly in the group of non-responders to systemic treatment. Three of the previously mentioned studies refer to postoperative complications or difficulty in post-surgical recovery. In all of them, the number and severity of the complications occurred did not suffer a substantial variation with respect to the expected complications, although these data are difficult to assess since none of the trials has a control group.

Pool analysis of data from all trials testing TKIs in the adjuvant setting demonstrated no benefit in terms of disease-free or overall survival for those patients M0 if surgical treatment achieved complete resection of neoplastic tissue. Adjuvant treatment should be recommended in the context of a multimodal protocol in the M1 patient, in whom metastatic spread cannot be completely controlled with local treatment. The recommendations for this type of patient are the same as for any patient with metastatic disease (M+) and should be carried out according to risk stratification.

## Figures and Tables

**Figure 1 biomedicines-11-00204-f001:**
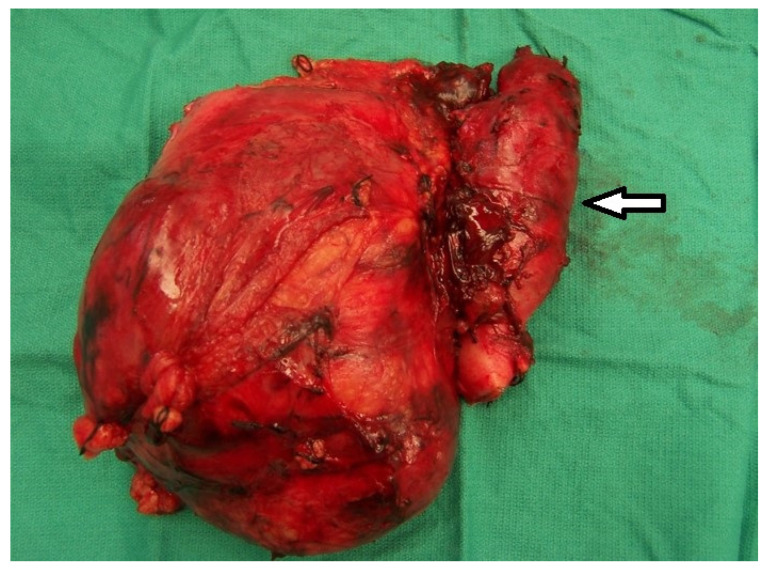
En-bloc resection of large right renal cell carcinoma with inferior vena cava containing tumor thrombus inside (white arrow).

**Figure 2 biomedicines-11-00204-f002:**
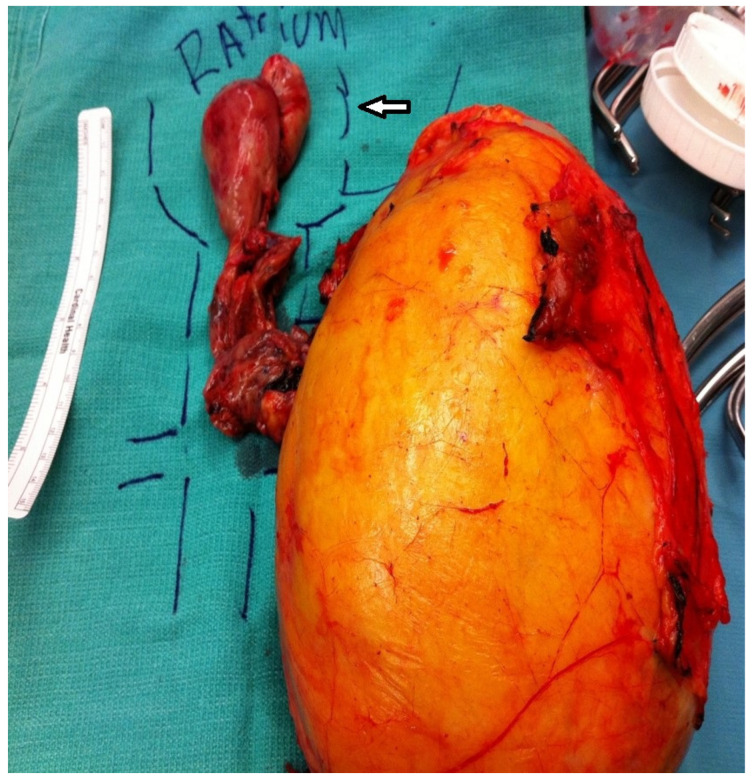
Surgical specimen of a large left renal cell carcinoma with a large tumor thrombus extending into the right atrium (white arrow).

**Table 1 biomedicines-11-00204-t001:** Published trials in neoadjuvant therapy in locally advanced renal cell carcinoma.

Author, Year	Patients (*n*)	Agent	Response Rate	Toxicity Rate (Grade 3–4)
Thomas et al. [52], 2009	*n* = 19	sunitinib	16%	37%
Silberstein et al. [53], 2010	*n* = 14	sunitinib	29%	21%
Hellenthal et al. [57], 2010	*n* = 20	sunitinib	85%	30%
Rini et al. [58], 2012	*n* = 28	sunitinib	80%	64%
Cowey et al. [59], 2010	*n* = 30	sorafenib	7.1%	9.0%
Hatiboglu et al. [60], 2017	*n* = 12	sorafenib	29%	50%
Karam et al. [61], 2014	*n* = 24	axitinib	46%	8.0%
Lane et al. [62], 2015	*n* = 72	sunitinib	59%	14%
McDonald et al. [54], 2018	*n* = 47	sunitinib	34%	30%
Rini et al. [64], 2015	*n* = 25	pazopanib	92%	64%
Cost et al. [65], 2011	*n* = 25	sunitinib	44%	Not reported
Bigot et al. [66], 2013	*n* = 14	sunitinib or sorafenib	57%	21%
Zhang et al. [67], 2015	*n* = 18	sorafenib	22%	11%
Field et al. [68], 2019	*n* = 53	sunitinib	28%	74% (median grade 1, grade 3 not reported)

## Data Availability

Not applicable.

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
