# Peer review of "Current Approaches in Surgical and Immunotherapy-Based Management of Renal Cell Carcinoma with Tumor Thrombus"

_biomedicines, 2023, doi:10.3390/biomedicines11010204_

Round 1

Reviewer 1 Report

This review is to provide a detailed description of specific surgical approaches and techniques used to minimize intra- and postoperative complications during radical nephrectomy and tumor thrombectomy of Renal cell carcinoma with tumor thrombus extension and to discuss the major developments in the neoadjuvant and adjuvant immunotherapy-based treatment of advanced Renal cell carcinoma. The manuscript is well-structured and well-discussed. However, some points should be checked and corrected before it's accepted in this journal. 

Therefore, according to my comments, I recommended the publication of the paper after minor revision.

[1]   The abstract need more clarity.

[2]   The study's background should be clearly stated. Describe the introduction and review of the work (Please add more information).

Author Response

[1]   The abstract need more clarity.

Thank you for your comments. We have revised the abstract to better fit our review’s background (lines 15-28). 

[2]   The study's background should be clearly stated. Describe the introduction and review of the work (Please add more information).

Thank you for your comments. We have revised the introduction to better described the background of our review (lines 112-117). 

Reviewer 2 Report

Dear authors,

your paper is a very well written and interesting report on the neoadjuvant management of locally advanced renal cell carcinoma. 

Yet some modification should be made:

1) the authors should add a table summarizing all the systemic therapy studies cited into the paper, with one column indicating the paper, the second column in wich the drug used is named, a third column with the response rate and a last column with toxicity rate-

2) Although I acknowledge that few evidence is available regarding the use of ICI in the neoadjuvant setting for locally advanced renal cell carcinoma, i think that some evidence may derive from clinical trials for metastatic renal cell carcinoma. THerefore, paper such as PMID 34750035 or 34103181 should be discussed in paragraph 3.1.5 .

3) the authors must cite into paragraph  all the papers regarding adjuvant treatment of RCC such as keynote 564. checkmate 914, IMmotion-010 even in the form of a press release.

4) there are a few typos in the text that should be checked. 

I acknowledge that i asked for a few revisions,  but i think that those revision could transform a good paper into an excellent one.

Author Response

1) the authors should add a table summarizing all the systemic therapy studies cited into the paper, with one column indicating the paper, the second column in wich the drug used is named, a third column with the response rate and a last column with toxicity rate-

Thank you for the comments are suggestions. We agree that a table would nicely summarize the trials cited into the paper. We have added Table 1: Published Trials in Neoadjuvant Therapy in Locally Advanced Renal Cell Carcinoma (line 546).

2) Although I acknowledge that few evidence is available regarding the use of ICI in the neoadjuvant setting for locally advanced renal cell carcinoma, i think that some evidence may derive from clinical trials for metastatic renal cell carcinoma. THerefore, paper such as PMID 34750035 or 34103181 should be discussed in paragraph 3.1.5 .

Thank you for the suggestion. We have included a discussion of a post-hoc analysis of Checkmate 214 into section 3.1.5 (lines 594-602) as follows: 

“A post-hoc analysis of CheckMate 214 trial [73], nivolumab nivolumab plus ipilimumab, demonstrated superior efficacy over sunitinib in patients with previously untreated International Metastatic RCC Database Consortium (IMDC) intermediate/poor-risk advanced RCC. The trial showed improved overall survival and progression-free survival benefits with nivolumab plus ipilimumab (N=60) vs. sunitinib (N=52) in intermediate/poor-risk patients with sarcomatoid features. A higher proportion of patients achieved an objective response (56.7% vs 19.2%) and a higher proportion of patients achieved a complete response (18.3% vs 0%).”

3) the authors must cite into paragraph  all the papers regarding adjuvant treatment of RCC such as keynote 564. checkmate 914, IMmotion-010 even in the form of a press release.

Thank you for the suggestion. We have cited the mentioned trials into section 3.1.5 (lines 589-590). 

4) there are a few typos in the text that should be checked. 

Thank you. We have reviewed the manuscript in its entirety and have made the necessary grammatical corrections.